# Genotyping of *Le* and *Af* Haplotypes in Dry Pea (*Pisum sativam* L.) with Field Trials: Short and Semi-Leafless Plants Are Not Always Better in Kazakhstan

**DOI:** 10.3390/plants14223479

**Published:** 2025-11-14

**Authors:** Bauyrzhan Arinov, Gulmira Khassanova, Aray Zailasheva, Marzhan Kuzbakova, Kazhymurat Mussynov, Tatyana Sereda, Assemgul Kipshakbayeva, Satyvaldy Jatayev, Karakoz Tolenova, Crystal Sweetman, Colin L. D. Jenkins, Kathleen L. Soole, Yuri Shavrukov

**Affiliations:** 1Faculty of Agronomy, S. Seifullin Kazakh AgroTechnical Research University, Astana 010000, Kazakhstan; arinov_1982@mail.ru (B.A.); azailashova@gmail.com (A.Z.); happy.end777@mail.ru (M.K.); kazeke1963@mail.ru (K.M.); kipas78@mail.ru (A.K.); s.jatayev@kazatu.edu.kz (S.J.); 2A.F. Khristenko Karaganda Agricultural Experiment Station, Karaganda Region 100435, Kazakhstan; sereda_t@bk.ru; 3Faculty of Medicine and Healthcare, Al-Farabi Kazakh National University, Almaty 050040, Kazakhstan; tolenova.kd@gmail.com; 4College of Science and Engineering, Biological Sciences, Flinders University, Adelaide, SA 5042, Australia; crystal.sweetman@flinders.edu.au (C.S.); colin.jenkins@flinders.edu.au (C.L.D.J.); kathleen.soole@flinders.edu.au (K.L.S.)

**Keywords:** *Afila*, ASQ molecular markers, drought, genotyping, haplotype, *Le* gene, lodging, pea, regular (conventional) leaves

## Abstract

Lodging of pea plants has long been one of the greatest problems encountered at harvest, but it can be avoided with reduced plant height (PH) and increased number of tendrils. Two genes, *Le* (stem length), Psat5g299720, for PH, and *Af* (*Afila*), Psat2g173360, for semi-leafless types with multiple tendrils, were studied in the 6 most popular pea cultivars in Kazakhstan and in 60 pea accessions from a germplasm collection. ASQ molecular markers were developed based on an identified SNP in the *Le* gene. Two groups of 41 tall and 25 short (semi-dwarf) pea plants with average PHs of 83.6 and 56.3 cm, respectively, showed dominant and recessive *Le* alleles. Nine haplotypes of *Af* gene were found in the same set of pea genotypes and the *PsPALM1b* (Psat2g173360) gene was present in 48 pea plants with regular-type leaves, but it was absent (deleted) in 18 *afila*-type peas. Seed yields were assessed in the six major pea cultivars in field trials in the Akmola and Karaganda regions of Northern and Central Kazakhstan, respectively, during 2024 and 2025. The short and semi-leafless pea genotypes showed better results in favourable and wet conditions in the Akmola region, whereas tall pea genotypes with regular leaf types were more productive in drought conditions, realising their potential in the Karaganda region. The results for 60 pea accessions in the same regions in 2025 followed a similar trend. Finally, we concluded that drought can influence the significance of the lodging problem in pea genotypes in Kazakhstan, where an earlier start and faster growth of taller plants with regular leaves can become much more important traits for better drought tolerance and seed production.

## 1. Introduction

Garden pea (*Pisum sativum* L.) is the classical plant species for genetic analysis, since it was first used by Gregor Mendel for his ‘Inheritance laws’ [1,2]. However, Mendel’s favoured plant species is now of great importance in our current era of genomics and genotyping, where molecular markers and marker-assisted selection are at the forefront of modern pea breeding [3].

‘Field’ or ‘dry’ pea is the same species, differing only in the time of harvesting and purpose of use. The field pea is a very important legume crop for food and animal feeding [4]. One of the biggest problems in crop production is a high level of plant lodging [5], especially in legume and pea plants [6]. In wet seasons with high humidity, the lodging problem is further compounded by stronger disease infection rates [7].

Lodging can be averted by reduced plant height (PH), changes in tendril number, increased stem strength, and stronger standing ability of plants, including in legume species and pea [8,9]. Resistance to lodging has long been studied using QTL analysis with different approaches in various crops [10], including pea [11,12]. For example, in three hybrid populations of pea in Canada, major QTLs for lodging resistance were found in LG3a and LG3c chromosomes, with phenotyping variances in the range 28.5–35.3%. These overlapped with the QTLs for PH [13].

Plant height was shown to be directly involved in resistance to lodging, especially during the ‘Green revolution’ era of breeding in wheat and other cereals [14] as well as in *Brassica* species [15]. Semi-dwarf wheat mutants have reduced PH, with shorter and thicker straws, which mechanically prevent plant lodging [16]. Stems of legume plant species are very different from cereals and may be thick and strong as in soybean and faba bean or weak as in pea. Additionally, stem diameter but not stem wall thickness was the most critical trait for lodging in pea, since stems are always thinner in the lower part of the stem compared to the middle or upper part [17]. Therefore, resistance of semi-dwarf plants to lodging with shorter stems and PHs is related to the centre of biomass being located closer to roots, and not only the mechanical property of their stems [18]. There are many reports indicating that shorter pea plants with lower PHs showed better lodging resistance [17,19].

Gibberellic acid (GA) is a well-known plant hormone and mutations in genes controlling the biosynthesis and metabolism of GA showed extreme effects, including a semi-dwarf PH and significantly shorter stems in plants of various species [20]. One of the earliest genes described in the Mendelian garden pea was the *Le* gene (stem length) encoding gibberellin 3β-hydroxylase (GA3ox). *Le* was studied and cloned in pea [21,22] and its orthologs in *Medicago truncatula* [23]. The overexpression of *PsGA3ox1* transgene in semi-dwarf lines of pea resulted in a restored level of GA and increased stem length and PH in transgenic pea plants [24]. There are many other genes controlling PH in addition to *Le* but most of them belong to GA-related enzymes and metabolite compounds and, therefore, are linked together in the one biochemical pathway of GA [25,26].

Nevertheless, PH and stem length are not the only traits important for resistance to lodging in pea. Tendrils, as modified parts of leaves, can have an even stronger impact on making pea plants non-lodging [27,28]. Tendrils can help pea plants to better support each other by binding all plants together to form a massive pea community connected and supported by their neighbours through their tendrils. Such ‘tendril-connected’ taller-growing plants have improved air movement between plants, leading to more uniform ripening and drying of pods and reduced development of fungal diseases.

In general, the regular (or conventional) leaf type in pea plants comprises several parts: one basal pair of stipules, several proximal leaflet pairs with following distal tendril pairs, and one last terminal tendril. It is also obvious that more tendrils can provide better and stronger connections between pea plants, supporting their upright growth. However, tendrils occur only from modified leaflets [29] as illustrated using primary and secondary veins [27]. Therefore, producing more tendrils will result in a reduced number of leaflets.

There are several known genes controlling the development of stipules and transformation of leaflets into tendrils. The *Unifoliata* (*Uni*) gene is involved in the development of simple, bifoliate, or trifoliate leaves without tendrils, and it was reported as promoting high levels of auxin in pea leaves [30,31]. Mutations in the *Leaflet development* (*Lld*) gene cause abortion of leaflets and tendrils [32].

The semi-leafless mutation *afila* (*af*) is one of the earliest known and is well-studied in the corresponding gene *Af*, which was recently identified as two genes *Palmate-like pentafoliata1*, *PsPALM1a* (Psat2g173880), and *PsPALM1b* (Psat2g173360), closely located in the genetic fragment of chr.2, and encoding a C2H2 zinc finger transcription factor [33,34]. As well as natural spontaneously occurring mutations, novel *afila*-type mutants were generated via X-ray treatment in the background of two dry pea cultivars [35].

Examples of improved upright-growing pea plants were shown with an introgression of *afila* genotype backcrosses (BCs) in tropical garden pea. The semi-leafless BC lines with more tendrils did not need vine staking, and their weight of fresh seeds was similar to the recurrent parent with regular leaves and fewer tendrils [36]. Additionally, semi-leafless *afila* pea genotypes, associated with upright stem growth and less lodging, were reported to have cooler canopies, a shorter vegetative period, earlier flowering, and reduced reproductive abortion during heat stress in field trials in Canada [37].

However, the introgression of the recessive allele *af* encoding the semi-leafless phenotype of pea can have potential drawbacks compared to traditional leafed pea accessions. Due to the reduction in leaf area and chlorophyll content in *afila* genotypes, there can be up to a 1.5-fold decrease in photosynthesis in stipules and fewer small leaflets compared to pea with regular leaves [38]. This can result in less biomass production, and it was reported to reduce seed yield by an average of 10% in *afila* sibling lines compared to those with regular leaves in all 12 studied hybrid populations, possibly due to a shorter period of plant growth and flowering [39], whereas no such differences were reported in two Russian cultivars of pea [38]. In a mixed sowing of pairs of pea genotypes, near-isogenic lines (NILs) with the traditional leafy type and semi-leafless *afila* type showed better lodging resistance and seed yield compared to pure NILs with leafy types, whereas no such difference or benefit was reported compared to pure *afila* NILs [40].

Regarding seed yield, regular-leafed pea plants have higher yield potential, but this can be reduced dramatically after lodging. In contrast, semi-leafless pea genotypes, in general, have smaller yield potential, but it can be compensated by better resistance to lodging compared to conventional pea accessions with full leaves [29]. Very different results were presented in a more recent study of 24 regular leaf and 30 semi-leafless pea genotypes grown in three environments in Germany [41]. The authors reported that on average, for all semi-leafless pea accessions, their seed yield and plant biomass were higher by 51% and 40%, respectively, compared to regular-leafed peas. However, it was emphasised that many of the normal-leafed pea genotypes were old and showed extremely high lodging scores of 7.9 compared to a 2.4 score in semi-leafless peas (on a 10-unit scale). Therefore, the reason for the advantage in seed yield can be directly related to the multiple tendril trait helping to promote more upright-growing plants and better resistance to lodging in semi-leafed pea genotypes in the study [41].

It was shown that both genes, *Le*/*le* and *Af*/*af*, controlling plant height and semi-leafless tendril plants, together cover the majority of the phenotype variation for lodging in field pea of a recombinant inbred line population [12]. Therefore, both genes are very important for the development of non-lodging pea genotypes. However, pea breeders must pay greater attention to the potential disadvantages of the introgression of these genes. There is not only the possible seed penalty from the shorter vegetative period as mentioned above but also the risk of reduced height to first pod in semi-dwarf *le* pea genotypes, as was shown recently in pea and other legumes [42].

Nevertheless, it was stated that all modern dry pea cultivars have a shortened stem (<90 cm) and the *afila* leaf type [17]. This may be an ‘overgeneralisation’ since some (albeit not many) currently grown pea cultivars and breeding lines have medium–high PH and normal leaves with regular tendrils. For example, only 43% were reported as having semi-leaflet *afila* among 147 cultivars of U.S. garden pea registered over a recent 20-year period (1990–2010) [43].

Abiotic stresses like drought and heat affect pea plants, especially in the flowering and seed-developing stages, regardless of PH and leaf type. However, if drought occurs in the early stages of pea plant development, semi-dwarf genotypes can be affected much more strongly and, in this case, there is no lodgement at all or it plays only a minimal role in seed production, as was shown in the example of lentil [44].

The ‘escape strategy’ was proposed with more rapid growth of semi-leafless pea accessions or other plant species, but this can be beneficial only if the onset of heat or drought stress is terminal [45,46]. However, earlier flowering *afila* pea genotypes can lose all the advantages of their tendrils and upright growth if drought occurs earlier in the season and overlaps with flowering, which very often happens in Kazakhstan and other Central Asian countries, based on the example of wheat [47].

In dry continental climates such as Kazakhstan, early-season droughts often coincide with flowering, potentially cancelling the agronomic advantages of semi-leafless and semi-dwarf pea genotypes. While these genotypes offer improved lodging resistance and cooling under moist conditions, their performance under early drought stress remains unclear. The literature reports are conflicting regarding whether these traits consistently enhance yield under variable environmental conditions. There is insufficient information on how *Le* and *Af* genotypes affect pea growth and yield in early-season drought-prone environments. It is unclear whether their benefits in lodging resistance and upright growth are sustained when drought occurs during critical developmental stages.

We hypothesise that the agronomic performance, e.g., yield and lodging resistance, of pea genotypes with different *Le* and *Af* haplotypes is significantly affected by early-season drought conditions in Kazakhstan.

The aim of this study was to evaluate the influence of *Le* and *Af* haplotypes on the phenotypes and agronomic traits of different pea accessions, using molecular genotyping with allele-specific quantitative PCR (ASQ) and PCR-based markers, and to assess phenotypes in field trials in environmental conditions in Kazakhstan.

## 2. Materials and Methods

### 2.1. Plant Material and Trait Evaluations

Seeds of field pea (*Pisum sativum* L.) accessions were provided by S.Seifullin Kazakh AgroTechnical Research University (KATRU), Astana (Kazakhstan), and Australian Grains Genebank, Horsham (Australia) [48]. Examples of four pea accessions grown in pots with soil that show the different growth habits are present in Figure 1. The full list of pea germplasms studied is provided in Appendix A.

Six major pea cultivars grown locally in Kazakhstan included (in alphabetical order) the following: (1) Aksaiskiy usaty-55 (Russia, 2011), PH = 85–95 cm, semi-leafless, 71–73 days to maturity [49]; (2) Astronaute (NPZ, Germany, 2013), PH = 46–80 cm, semi-leafless, 76–84 days to maturity [50]; (3) KASIB (Kazakhstan, 2015), PH = 60–90 cm, semi-leafless, 65–86 days to maturity [51]; (4) Omskiy neosip. (Russia, 1993), PH = 60–130 cm, conventional leafy, 56–98 days to maturity [52]; (5) Oris (Өris) (Kazakhstan, 2020), PH = 60–85 cm, conventional leafy, 80–107 days to maturity [53]; and (6) Status (Kazakhstan, 2017), PH = 50–75 cm, semi-leafless, 69–76 days to maturity [54].

For molecular experiments, seeds were sown in 20 cm diameter pots with 3.0 kg of soil mix comprising equal volumes of commercial potting mix and soil from a research field near KATRU (Kazakhstan). Plants were grown at KATRU Campus during the summer season on the same time schedule as the research field plots. Pots were watered twice weekly on a portable scale, keeping the soil moisture level consistent at 80% field capacity.

For field experiments, two trial types were used. Plants of six major pea cultivars were grown in research fields in Akmola and Karaganda regions over two years (2024–2025) in six row plots of 1 m^2^ and 80 plants, with 8 cm between plants in a row and 20 cm between rows. Plants of each of the 60 pea accessions from the germplasm collection were tested in the same fields and in both locations, in 2025 only, in a single row plot, 1 m in length and containing 12 plants, with the same plant density. Each genotype of major pea cultivars and accessions had three replicated plots with a completely randomised plot design.

Data on precipitation were retrieved from the local meteorological stations for the first and second parts of the pea growing season, May–June and June–July, respectively, and this is graphically presented in Figure 2.

In the Akmola region (Northern Kazakhstan), 2024 was considered as having moderate drought in the beginning and being favourable in the second part, whereas 2025 was classed as having moderate drought in the beginning and highly excessive rainfall in the second part of plant growth. In the Karaganda region (Central Kazakhstan), 2024 was considered as having strong drought in the beginning and being favourable in the second part, and 2025 was classed as having moderate drought in the beginning and strong drought in the second part of plant growing (Figure 2).

### 2.2. Phenotyping

Plant height was measured as centimetres from the soil surface to the apical meristem of the longest stem, ignoring tendrils at the end of the flowering stage, and calculated as the average from 10 plants in each genotype. The measurement was repeated at post-harvest analysis, confirming the accuracy of the PH results.

For leaf type, the conventional leaves with terminal tendrils or the semi-leafless ‘*afila*’-type with multiple extra tendrils were observed and photographed at the flowering stage and checked at post-harvest analysis for the absence of aberrant plants in each genotype, following the descriptions and photos in earlier published papers [33,34]. Examples of four pea accessions with contrasting PHs and leaf types are shown in Figure 1 above.

To determine the lodging index (LI), groups of 10 plants from each genotype grown in field trials were estimated for resistance to lodging or non-lodging capacity, ranging from ‘Class 1’, completely lodging plants, to ‘Class 5’, non-lodging plants with full resistance to lodging based on the traditional scoring system used in other legumes [44], as well as one more specific for pea [55].

For seed yield, seeds were harvested in each plot and weighed after threshing and cleaning. For all pea genotypes, seed yield was calculated as the average of three plot replicates with further statistical treatment. For six major pea cultivars, seed weight was used directly from 1 m^2^ plots, whereas seed yield for 60 pea accessions in smaller plots was re-calculated from average to 1 m^2^ plots, making it suitable for the comparison. Average seed yield and standard errors were calculated for each genotype with further statistical treatment.

Relative yield class (RYC) was based on ‘crop yield index’, originally described in [56], and more recently on ‘yield class’ [57]. The simplified ratio was used as follows:$$\mathrm{RYC}\;=\;Y_{\mathrm{genotype}}/Y_{\mathrm{average}}$$where Y_genotype_ is data for yield in a certain genotype and Y_genotype_ is average yield for all studied genotypes in the field trial, with a following *t*-test conducted for the significant difference between each genotype and the average for all studied. The rationale of RYC use is the simple presentation of yield class for the comparison between different pea accessions studied in each field trial. The ranking of studied genotypes into three classes is very simple based on significant RYC values, as follows: RYC > 1 = high class (H); RYC < 1 = low class (L); and RYC = 1 (no significant differences) = medium class (M) of RYC.

### 2.3. Molecular Genetic Analysis DNA Extraction

Molecular genetic analyses were carried out in parallel in both partner laboratories, at KATRU, Astana (Kazakhstan), and at Flinders University, Adelaide (Australia). Two young leaves with basal stipules from each plant were collected as one sample in 10 mL tubes and frozen in liquid nitrogen. DNA was extracted using either the CTAB method at KATRU, as described earlier [58] with minor modifications [42], or the phenol-chloroform method based on the initial protocol [59] with further modifications [60]. Frozen leaf samples were ground with ball bearings and a vortex mixer, ensuring samples remained frozen. The washed and dried DNA pellet was finally dissolved in 100 µL of 1/10-diluted TE Buffer with 25 µg of Ambion RNase Cocktail added (Invitrogen-Thermo Fisher Scientific, Waltham, MA, USA). The DNA concentration was measured by Nano-Drop spectrophotometer (Thermo Fisher Scientific, Waltham, MA, USA), and the DNA quality was assessed on a 1% agarose gel.

### 2.4. Le Gene: Sequencing, SNP Identification, ASQ Marker Development, and SNP Genotyping

The sequence of the *Le* gene, Psat5g299720 (LOC127087529), encoding Gibberellin-3 beta-hydroxylase, was retrieved from the Pulse database (PDB) [61], and the primers were designed based on the DNA sequence of the *Pisum sativum* reference genome of cv. Cameor. Primers for sequencing were designed targeting a fragment of 995 bp in the second exon and 3′-UTR, and their sequences are present in Appendix A.

Six major pea cultivars, described in Section 2.1, were used for sequencing. PCR was performed in 60 µL volume reactions containing 6 µL of template DNA adjusted to 50 ng/mL, and with the following components in their final concentrations as listed: 1 × PCR buffer, 2 mM MgCl_2_, 0.2 mM each of dNTPs, 0.25 mM of each primer, and 4.0 units of GoTaq Flexi DNA polymerase (Promega, Madison, WI, USA) in each reaction. PCR was carried out on a SimpliAmp Cycler (Thermo Fisher Scientific, Waltham, MA, USA) at KATRU and on a T100 Thermal Cycler (BioRad, Hercules, CA, USA) at Flinders University, using a programme with the following steps: initial denaturation, 94 °C for 2 min; 35 cycles of 94 °C for 10 s, 56 °C for 15 s, and 72 °C for 1 min; and a final extension of 72 °C for 3 min. PCR visualisation was performed using 1% agarose gel.

The PCR products were purified using either a PCR Purification kit (Syntol, Moscow, Russia) at KATRU or FavorPrep PCR Purification kit (Favorgene Biotec Corp., Taiwan, China) following the corresponding manufacturer’s protocols, and their concentrations were measured using NanoDrop spectrophotometer (Thermo Fisher Scientific, Waltham, MA, USA). Sanger sequencing was carried out using BigDye Terminator v3.1 reagents and SeqStudio at KATRU or using AB-3730 Genetic Analyzer, both from Applied Biosystems (Thermo Fisher Scientific, Waltham, MA, USA), at the Australian Genome Research Facility (Adelaide, Australia). SNPs were identified using manual comparison of the visualised sequences using the Chromas computer software programme, version 2.0. The identified SNPs were verified by sequencing of the same PCR products in both directions.

Plant genotyping was based on the ASQ method from the original protocol [62] with the following modifications [63]: The molecular probe with a short 4 bp tag was used, and two allele-specific forward primers together with one reverse primer were designed for the identified Psat5g299720-SNP1. The composition of the PCR cocktail for ASQ genotyping in a total volume of 10 µL and sequences of the allele-specific primers and universal molecular probes are presented in Appendix A.

The primers and molecular probes were obtained from DNA Synthesis (Moscow, Russia) for KATRU and purchased from Sigma-Merck (Saint Lois, MD, USA) for Flinders University. Reactions were carried out in a 96-well microplate. Allele discrimination was determined using a QuantStudio-7 Real-Time PCR system instrument (Thermo Fisher Scientific, Waltham, MA, USA) at KATRU and CFX96 Real-Time qPCR system with CFX Maestro software, version 2.3 (BioRad, Hercules, CA, USA) at Flinders University, with automatically recorded fluorescence in both instruments. Amplification of FAM and HEX was checked and controlled, whereas SNP calling and genotyping results were determined in a post-run step with analysis of Real-Time dRn setting. Genotyping experiments were carried out with three individual plants (biological replicates) and results were validated with two repeated runs (technical replicates) for each pea genotype. The accuracy was confirmed using ‘no template controls (NTCs)’ with sterile water instead of template DNA. All six major pea cultivars sequenced in our experiment showing clear SNPs were used as reference genotypes for allele discrimination.

### 2.5. Af Gene: PCR and Haplotype Identification

For *afila*-type leafless analysis, four genes were studied as follows: two *Palmate-Like Pentafoliata1* genes, *PsPALM1a* (Psat2g173880) and *PsPALM1b* (Psat2g173360), encoding C2H2 zinc finger transcription factor, and two flanking genes, *PsSA-RNA* (Psat2g173920), putative small auxin-up RNA coding gene, and *PsNaOD1* (Psat2g173320), acetylornithine deacetylase [33,34]. Similarly to the *Le* gene described in Section 2.4 above, the sequences of these genes were retrieved from the Pulse database (PDB) [61] using the *Pisum sativum* reference genome of cv. Cameor. The sequences of the primers and their design in the genes are present in Appendix A. Regular PCR conditions as described in Section 2.4 above were used, with a proportional reduction in the total reaction volume to 15 μL, as presented in Appendix A.

### 2.6. Statistical Treatment

Means, standard errors, and significance levels were calculated using one-way ANOVA with the post hoc Tukey HSD test to identify statistical differences for plant phenotypes, including PH, leaf type, and seed yield among pea accessions with confidence intervals of at least *p* < 0.05 [64]; unpaired *t*-tests for differences between haplotypes and plant phenotypes [65]; and Fisher’s Exact Test for plant phenotypes and RYCs in two regions [66]. GGE biplot analysis was performed using the Global-centred (E + G + GE) option to study GxE interaction [67]. For field trials, three fully randomised plots were used as replicates for each accession. For molecular analyses, at least three biological replicates (individual plants) and two technical repeats (instrumental runs) were used for each genotype and experiment.

## 3. Results

### 3.1. SNP Identification, ASQ Marker Development, and Genotyping of Pea Accessions for Alleles of Le Gene

Sequencing analysis of six major pea cultivars revealed the presence of a single SNP in the *Le* gene Psat5g299720 with clear molecular differences in pea genotypes for PH in two groups: (1) tall and moderate and (2) dwarf and semi-dwarf. This is shown in the example of cvs. KASIB and Astronaute (Figure 3). Four out of six major sequenced pea genotypes, Aksaiskiy usaty-55, KASIB, Omskiy neosip., and Oris, showed a [G] allele in the position of SNP1, and they were all characterised by a higher PH, falling into the tall/medium group. In contrast, two pea genotypes, Astronaute and Status, with a semi-dwarf phenotype had an [A] allele in the same SNP1.

Additionally, based on the Pulse database [59], the reference genotype, cv. Cameor, also belongs to the same group of semi-dwarf peas, with the [A] allele of SNP1 in the *Le* gene. The identified SNP1 was non-synonymous and missense, encoding an amino acid Ala229Thr substitution.

ASQ molecular marker Psat5g299720-SNP1 was developed based on the identified SNP for screening of all pea accessions used in this study. The genotyping was based on the amplifications of fluorophores FAM and HEX in plants with different alleles of the *Le* gene, and allele discrimination showed a clear distribution of ‘*aa*’ and ‘*bb*’ genotypes (Figure 4). Finally, the genotyping of pea plants with the Psat5g299720-SNP1 molecular marker distinguished between two groups of genotypes, (1) dwarf and semi-dwarf (*aa* genotypes) and (2) tall and medium (*bb* genotypes), and these results are present in Appendix A.

### 3.2. Af Gene Haplotype Identification in Pea Accessions

To study *Af* locus, haplotypes were identified in two major *PsPALM1* genes controlling the transformation of conventional leaves into semi-leafless plants, together with two flanking genes, *PsSA-RNA* and *PsNaOD1*. For the first gene, *PsSA-RNA* (Psat2g173920), on the distal part of the genetic fragment, the size of the amplified PCR product was different among studied pea genotypes, whereas the haplotypes in the other three genes were determined based on presence or absence of amplification bands.

In the result of *Af*-genotyping, 9 haplotypes were identified in main pea cultivars grown in Kazakhstan and in the germplasm collection with 66 accessions in total (Figure 5). The first 4 haplotypes indicated in blue were found only among pea plants with conventional leaf types, whereas haplotypes 5–9 indicated in green were fully related to the semi-leafless phenotype and additional tendrils. The presented results clearly indicate that only the gene *PsPALM1b* (Psat2g173360) can solely control leaf type development, where *afila*-type semi-leafless pea plants occur with a mutated, defective, or deleted gene. Gene *PsPALM1a* (Psat2g173880) as well as both flanking genes does not affect changes in leaf types in pea plants.

An example of gel-based scoring of amplified PCR products of 24 pea accessions for haplotypes in four studied genes is present in Figure 5. The identification of the used pea accessions is matching in both Figure 5 and Figure 6.

The first six samples in Figure 6 represent the main pea cultivars widely grown in Kazakhstan, and they represent all different haplotypes of the *Af* locus. Haplotype 1 with the full set of all four studied genes is present in sample No. 4, cv. Omskiy neosip., whereas sample No. 6, cv. Oris, belongs to haplotype 2 with the mutated or deleted gene *PsPALM1a*. Both these accessions have conventional-type leaves. Four other main pea cultivars have a semi-leafless habit and are distributed among haplotypes as follows: sample No. 3, cv. Astronaute, representing haplotype 5 with mutated or deleted genes *PsPALM1a* and *PsPALM1b*; sample No. 1, cv. Aksaiskiy usaty-55 (haplotype 6), where two genes, *PsPALM1b* and *PsNaOD1*, are mutations or deleted; sample No. 2, cv. KASIB (haplotype 7), with three mutated genes and wild-type *PsNaOD1*; and sample No. 5, cv. Status (haplotype 8), which also has three mutated genes but gene *PsSA-RNA* remains unchanged (Figure 5).

Other studied genotypes are pea accessions selected from the germplasm collection and they are added in Figure 5 and Figure 6. In addition to those described above, haplotypes 3 and 4 show different combinations of present and absent genes, but the main gene *PsPALM1b* is always present in plants with conventional-type leaves. For *afila*-type plants, the last haplotype 9 is identified with all four studied genes as being either absent or mutated, and an example of this haplotype is found in pea cv. Kaspa (Figure 5 and Figure 6). The complete genotyping results for the *Af* gene in all 66 pea accessions are present in Appendix A.

### 3.3. Genotype and Phenotype Association Analysis

The summary results are present in Table 1 showing statistical analysis between genotyping of *Le* and *Af* genes and phenotyping of all 66 pea plants for their PH and leaf type, respectively. In Table 1A, the discrimination between *Le* genotypes using ASQ Psat5g299720-SNP1 showed 25 ‘*aa*’ (FAM) genotypes and 41 ‘*bb*’ (HEX) genotypes. The phenotyping of the same pea accessions revealed the average PH in the two corresponding groups as being 56.3 and 83.6 cm, respectively. The *t*-test for comparison between two groups showed very high significant differences between the two studied groups, indicating the high efficiency of the chosen molecular marker (Table 1A).

In Table 1B, the distribution of 66 pea genotypes for two groups of *Af* haplotypes included 48 WT *Afila* and 18 natural mutant *afila* genotypes. This was almost identical to phenotyping of the same pea accessions for regular and leafless phenotyping with only one mismatch (difference) in each of the two classes. Fisher’s Exact Test confirmed that they were very similar and that there were insignificant differences between *Afila* genotyping and phenotyping of leaf types in the studied 66 pea accessions (Table 1B). The presented results confirm the efficiency of the chosen molecular markers and haplotype identification.

### 3.4. Field Trial Assessment

Six major pea cultivars were tested in field trials in two locations, the Karaganda region, Central Kazakhstan, and the Akmola region, Northern Kazakhstan, in two seasons, 2024 and 2025. The results of seed yield and arranged relative yield class (RYC) together with PH and lodging index are present in Table 2.

Results of seed yield were best in the first trial in the Akmola region in 2024 with relatively favourable weather conditions (Table 2A), and RYC was based on statistical differences between six studied pea cultivars. The top-ranked two cultivars with high RYC, Oris and Status, had medium PH with regular leaves and short semi-leafless peas, respectively, and had minimal lodging. Medium RYC was found in two very contrasting cultivars: (1) short semi-leafless and non-lodging cv. Astronaute and (2) tall cv. Omskiy neosip. with regular leaves and moderate lodging. The low-ranking pea cultivars had tall and medium PH, both leafless and with moderate–high resistance to lodging.

In the second field trial in the Karaganda region in 2024, with stronger drought at the beginning of plant growing, all six pea genotypes were affected, which can be observed in the PH which was reduced by as much as half in some genotypes. Regarding seed yield, two cultivars, Oris and Omskiy neosip., did not show any significant reduction. In contrast, seed yield in two other cultivars, Status and Astronaute, was reduced by two-fold or more (Table 2A).

However, weather conditions in 2025 were much harsher, making it unfavourable for plants but interesting for this study (Table 2B). In the Akmola region, early drought at the beginning of the growing period turned to wet in the middle of season with a lot of rain toward the end. Both tall pea cultivars were lodged almost completely and subsequently recorded much poorer seed production. The best seed yield was recorded in two medium-height pea cultivars, where cv. Oris showed strong stability in both favourable and unfavourable conditions but cv. Aksaiskiy usaty ‘jumped’ from ‘outsiders’ in 2024 to ‘champions’ in 2025. A short cv. Status lost ground from being high rank, whereas the second short cv. Astronaute remained unchanged in the middle ranks.

The extreme weather conditions in the Karaganda region in 2025 started with typical moderate drought but continued for longer when rains arrived too late in the growing season. The seed yield was low but both tall cultivars realised their potential and produced more seed yield compared to the other studied cultivars. In contrast, the smallest seed yield was recorded in short cultivars with a three-fold-lower seed yield compared to tall genotypes (Table 2B). To demonstrate simpler differences between tall and short genotypes, the summary results for seed yield and RYC in field trials with indicated weather conditions are shown in Figure 7.

The biggest difference in RYC and PH was observed between geographic locations rather than years and it was specifically related to the level of drought in the beginning and in the second part of pea plants’ growth. The short, semi-dwarf pea genotypes showed better results in the more favourable wet conditions and milder drought in the Akmola region, whereas tall pea genotypes were more productive in drought conditions, realising their potential in the Karaganda region (Figure 7).

The relationships between environments and genotypes were studied using GGE biplot (Figure 8). The conditions in Akmola-2024 and Karaganda-2024 were different but still more favourable for all six major pea cultivars studied. In contrast, conditions in Akmola-2025 and Karaganda-2025 were extreme, positioned a great distance from all studied genotypes.

Two cultivars, Status and Astronaute, both short and semi-dwarf, were located closer to the Akmola-2024 vector. Two other cultivars, Omskiy neosip. and Oris, tall and medium-sized, respectively, were very close to the vector of Karaganda-2024. The last two cultivars, KASIB (tall) and Aksaiskiy usaty (medium PH), were in the middle of the biplot, close to the central point (Figure 8). These GGE biplot results were very similar to those presented in Table 2 and Figure 7. The only difference was the scoring in 2025, with extreme conditions and very low seed yields.

To verify the results received from field trials with six major pea cultivars, similar field experiments were carried out with 60 pea accessions from the international germplasm collection, where 38 genotypes were tall and medium-height and with regular leaves and the remaining 22 genotypes were short with both leaf types (Table 3). Most tall plants showed a medium–low seed yield rank in the Akmola region, but this changed to a high RYC rank in the Karaganda region. Pea plants with medium PH and regular leaves had better RYC performance in the Akmola region but dropped to a medium RYC rank in the Karaganda region. However, statistically it was not significant due to the small number of studied genotypes. In contrast, seed yield rank changed from medium to low in leafless short and medium-sized pea plants when they were grown in the Akmola and Karaganda regions, respectively (Table 3). The presented results for 60 pea accessions have the same trend as six major pea cultivars grown in Kazakhstan (Table 2).

## 4. Discussion

The yield of a pea crop is determined by two general components: potential yield of plants grown in field conditions and loss of seeds during harvesting. Obviously, the potential must be high with minimal seed loss during harvesting. However, for pea plants, the ability for lodging resistance represents a much more important trait to ensure that pods with seeds can be harvested. In lodged pea plants, seed yield can be greatly reduced, even to the point of complete loss, with an accompanying reduction in seed quality. Plant height and leaf type with more or fewer tendrils are the two major traits influencing lodging of pea plants [17,19,27,28].

For PH, two major groups of pea plants are mostly used for cultivation in Kazakhstan and other countries, including semi-dwarf and medium–tall height, representing about 50–80 and 80–100 cm, respectively. In the current study, six major pea cultivars grown in Kazakhstan were also designated as tall–medium and short (semi-dwarf) PH, described in Section 2.1 and in Appendix A. In contrast, dwarf (shorter than 50 cm) and very tall plants (higher than 1 m) are rarely used and are usually employed only as a germplasm collection for hybridisation and selection in further breeding programmes, which was found in 60 studied pea accessions from the international germplasm collection. This general description of pea PH refers to plants grown in conditions of sufficient moisture. Therefore, semi-dwarf pea genotypes are typically shorter than 80 cm, whereas medium–tall pea plants have a PH higher than 80 cm in well-watered conditions, which was also confirmed in the current study.

Following our working hypothesis, the preferred and popular short PH and semi-leafless pea accessions will have very different plant growth and seed yield in early- and strong-drought conditions in Kazakhstan, where their PH is shifted down by 50–60 cm or below. To complete this task, the *Le* and *Af* genotypes in studied pea accessions must be confirmed.

In the current study, genotyping for SNP1 in *Le* gene helps to clearly and rapidly identify genetic polymorphism in pea accessions, cultivars, breeding lines, and hybrids, and is very similar to studies published earlier [21,22]. Allele discrimination with the application of ASQ molecular marker Psat5g299720-SNP1 showed a strong association with two groups of genotypes: (1) dwarf and semi-dwarf and (2) medium-height and tall plants. Such genotyping is important for rapid screening of *Le* alleles in large numbers of studied pea plants.

A very different case was found for the *Af* gene, with large deletions in one or both of the genes *PsPALM1a* and *PsPALM1b*. Our results confirmed that all pea genotypes with conventional leaf types and with terminal tendrils had both native *PsPAML1a* and *1b* genes, and they were identified as WT [33,34]. In a similar manner, all *afila* semi-leafless genotypes of pea with multiple tendrils showed the deletion of the *PsPALM1b* gene. In contrast, presence or absence of *PsPALM1a* was variable, indicating that it was not strictly required in controlling the semi-leafless trait because this gene could be deleted in pea plants with any type of leaves. Both flanking genes, *PsSA-RNA* (Psat2g173920) on the distal side, encoding small auxin-up RNA, and *PsNaOD1* (Psat2g173320) on the proximal side, encoding acetylornithine deacetylase, were also variable. All studied genes showed variability among nine haplotypes, where five haplotypes related to semi-leafless plants and four haplotypes were plants with a regular leaf type. Our presented results about *Af* genotyping of 66 studied pea accessions were in complete consensus with previously published reports [33,34].

How can this knowledge on *Le* allele genotyping and haplotypes of *PsPALM1* genes help to identify pea genotypes more or less resistant to lodging to enhance seed yield without losing seed quality? Plant height and many other traits are very variable depending on growing conditions. Therefore, precise genotyping for *Le* alleles helps better identify and designate WT pea plants with tall PH and semi-dwarf plants with shorter PH. In a similar way, haplotypes of the *PsPALM1b* gene and the surrounding genetic region clearly show genetic polymorphism for leaf types and differences between WT and semi-leafless plants. Therefore, the developed molecular markers can be used in marker-assisted selection (MAS) for large-scale genotyping of segregating populations in pea plants to identify genotypes with required PH and leaf types in early stages of plant growth and development. An example of such genotyping results is present in the current study in six major pea cultivars popular in Kazakhstan and in 60 accessions from an international germplasm collection. Our presented results show very similar trends and conclusions about pea genotyping and haplotype identification [68], using genotyping by sequencing [69] and whole-genome sequencing, as reported earlier [70].

As discussed above, both PH and leaf type are the most important for lodging of pea plants and their potential seed yield loss [39,41,55]. Phenotyping of pea plants is strongly dependant on environmental conditions in the fields, and it has a very big impact on plant development, PH, lodging ability, and ultimately seed yield [71,72]. However, only a few published papers have been found describing pea accessions with variable PHs and leaf types grown in field trials under drought, indicating the gap in our knowledge that is addressed in the current study.

Early drought is typical for Kazakhstan, as a country with a strong continental climate [46], and it can affect pea plant growth from germination to the beginning of flowering. Later drought can cause the cessation of plant growth, whereas wet seasons increase lodging, especially of tall pea plants. This occurred in the current study in both geographic locations, Akmola and Karaganda regions, in the 2024 and 2025 growing seasons (Figure 2 and Figure 7).

In the current study, semi-leafless pea genotypes with shorter PHs performed better and were highly competitive in favourable or wet conditions, where their potential could be realised completely. This type of pea plant had perfect growth and high resistance to lodging, resulting in very high seed yield and no loss during harvesting. Similar results were reported earlier for peas grown in a Mediterranean environment, where shorter semi-leafless genotypes were identified as competitive or superior to other taller genotypes with regular leaves [55,73].

However, in conditions more typical in Kazakhstan, with moderate–high drought either in the first or second part of plant growth, only pea genotypes with medium and tall PHs were demonstrated to be more suitable and competitive for seed yield (Table 2 and Table 3 and Figure 7), including the GxE interaction presented in the GGE biplot (Figure 8). Our presented results were similar to those published earlier in pea plants grown across climatically contrasting environments [74,75]. Additionally, this statement was confirmed in a published report where semi-dwarf pea plants, after treatment with the hormones gibberellic acid (GA3) and brassinosteroid (BR), increased in PH by 50% and leaf area by 66%, and under drought demonstrated a 60% higher seed yield [76].

Fast early growth at the seedling and young plant stages can secure potential seed yield in conditions of later-onset drought, and this yield will obviously be reduced compared to well-watered conditions. Shorter pea genotypes with slower early growth are unable to be competitive in the drought conditions of Kazakhstan. Similar results with instability of yield in semi-leafless pea genotypes under drought were reported in Türkiye [75], showing consensus with the results for pea accessions grown in a dry environment. Therefore, the presented results confirmed that our working hypothesis was correct and that popular semi-dwarf and semi-leafless pea genotypes in early drought did not show any advantages over traditional tall pea plants with regular leaf types, confirmed with *Le* and *Af* genotyping.

In conclusion, in Kazakhstan, the major problem for pea genotypes is now their changes under drought and that the traits provided by short and semi-leafless-type plants are no longer favoured. There is no problem with lodging in conditions of moderate–high drought, but faster initial growth of taller plants with regular leaves and better drought tolerance are much more important traits to integrate into modern breeding programmes in pea.

## Figures and Tables

**Figure 1 plants-14-03479-f001:**
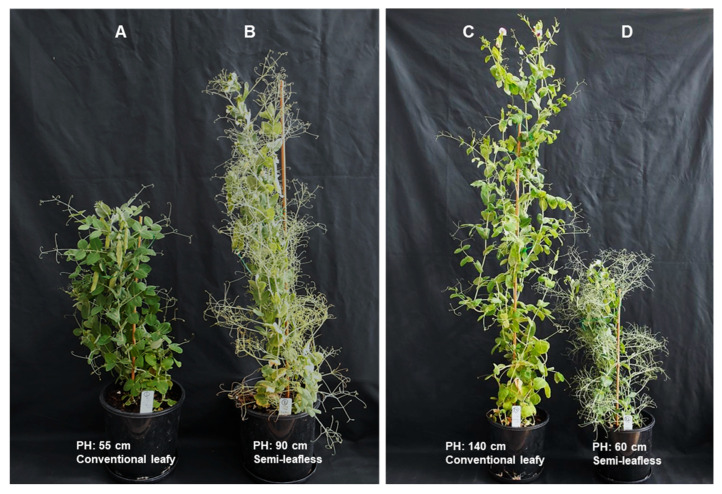
Images of pea plants from the germplasm collection grown in pots with soil. (**A**) cv. New Season (USA); (**B**) cv. Profi (Denmark); (**C**) accession Arakass (Hungary); and (**D**) breeding line 2002(10)-4 (China).

**Figure 2 plants-14-03479-f002:**
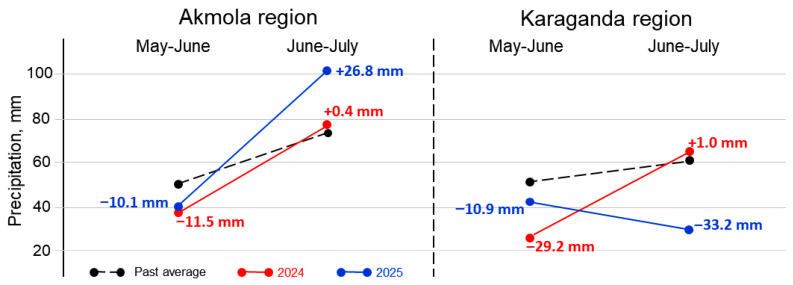
Precipitation data for growing seasons, May–June and June–July in 2024 and 2025, in Akmola and Karaganda regions (Kazakhstan). Black dots and dashed lines represent past average precipitation. Precipitation in 2024 and 2025 is shown in red and blue, respectively. Differences between actual precipitation and past average are indicated with either plus or minus in mm of precipitation, accordingly.

**Figure 3 plants-14-03479-f003:**
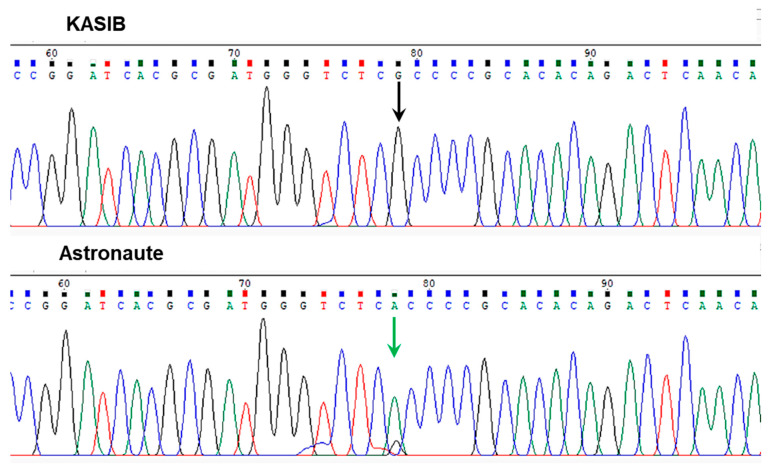
Fragments of Sanger sequencing of the second exon in the *Le* gene (Psat5g299720) in two pea accessions. Positions of the identified SNP are shown by arrows with different colours.

**Figure 4 plants-14-03479-f004:**
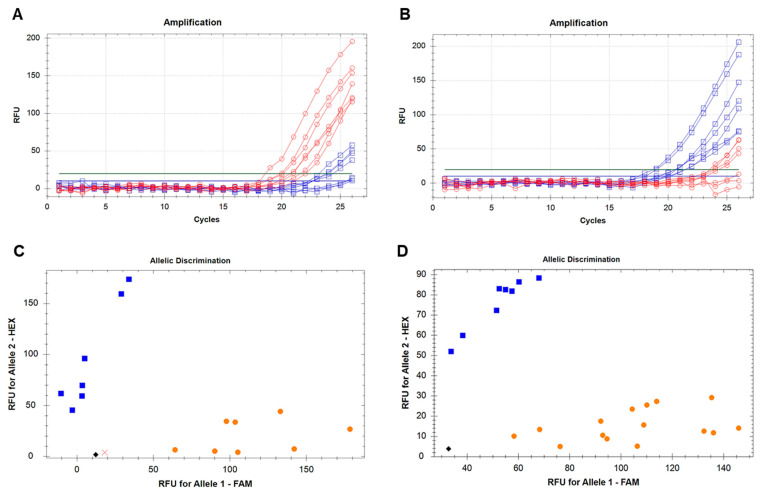
Development and analysis of ASQ molecular marker Psat5g299720-SNP1 for genotyping of pea accessions for *Le* gene. Amplification of fluorophore FAM, red lines (**A**), and HEX, blue lines (**B**), in genotypes with contrasting *Le* alleles. Discrimination between ‘*aa*’ and ‘*bb*’ genotypes, designated by orange and blue dots, respectively, in two rounds of genotyping (**C**,**D**). The black rhombus shows the no template controls (NTCs) using water instead of template DNA, whereas a cross indicates a genotype with undetermined status.

**Figure 5 plants-14-03479-f005:**
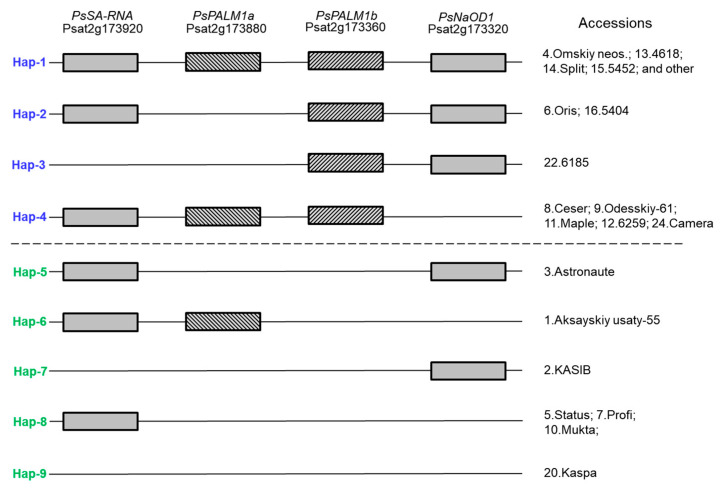
Schematic presentation of nine *Af* haplotypes with four studied genes, indicated on top of figure. Rectangles show presence of native (wild-type) genes, whereas absence of these genes can be related to their mutations or deletion. List of pea accessions with corresponding haplotypes is arranged according to results presented in Figure 6.

**Figure 6 plants-14-03479-f006:**
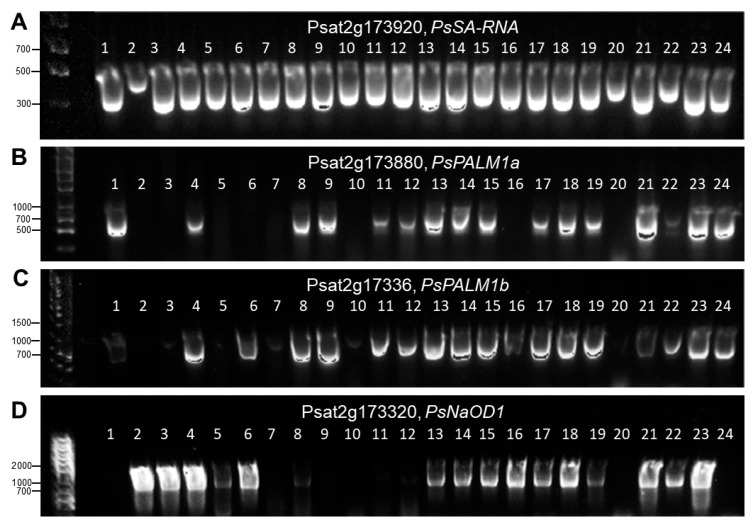
Images of presence or absence of PCR products and their separation by electrophoresis in 1% agarose gel. Results for four studied genes are present in separate panels (**A**–**D**). The order of the used 24 DNA samples remains unchanged in all four panels as follows: (1) Aksaiskiy usaty-55; (2) KASIB; (3) Astronaute; (4) Omskiy neosip.; (5) Status; (6) Oris; (7) 3832 Profi; (8) 4969 Ceser; (9) 5035 Odesskiy-61; (10) 5186 Mukta; (11) 5848 Maple; (12) 6259 IG-134626; (13) 4618 IFPI-5208; (14) 4859 Split; (15) 5452 IFPI-4701; (16) 5404 IFPI-4113; (17) 5452 IFPI-4701; (18) 5502 IFPI-4938; (19) 5529 IFPI-5155; (20) 6570 Kaspa; (21) 6167 IFPI-5160; (22) 6185 IFPI-5205; (23) 6622 L0358; and (24) 6604 Camera. Full details of the used pea accessions are present in Appendix A.

**Figure 7 plants-14-03479-f007:**
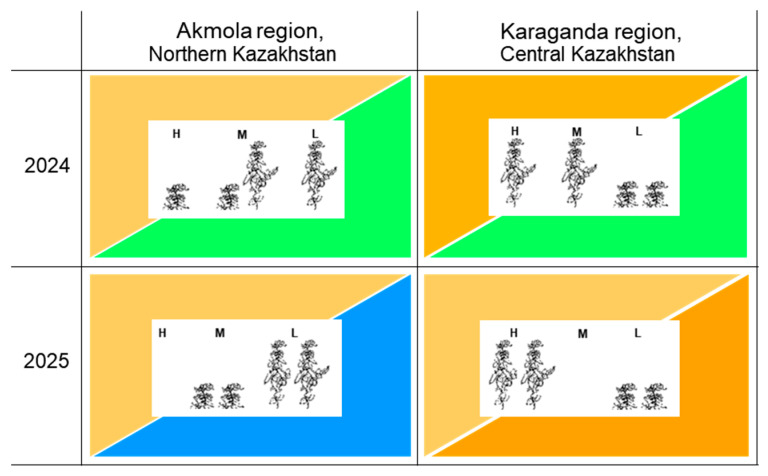
Schematic summary results of field trials in two regions of Kazakhstan and in two years with weather conditions detailed, where two tall and two semi-dwarf pea cultivars are included. Each field trial is represented by a rectangle split into two triangles. Top-left and bottom-right triangles indicate weather in the first and second half of pea plant growth, respectively (from sowing to start of flowering and from flowering to maturity). Brown colour relates to drought, green is optimal conditions, and blue is excessive rain. Darker colour represents stronger intensity of drought. Three relative yield class (RYC) evaluations are shown in centre of each trial and indicated as high (H), medium (M), and low (L). Short and tall plants in centre represent semi-dwarf cultivars Status and Astronaute and tall cultivars Omskiy neosip. and KASIB. Details are present in Table 2.

**Figure 8 plants-14-03479-f008:**
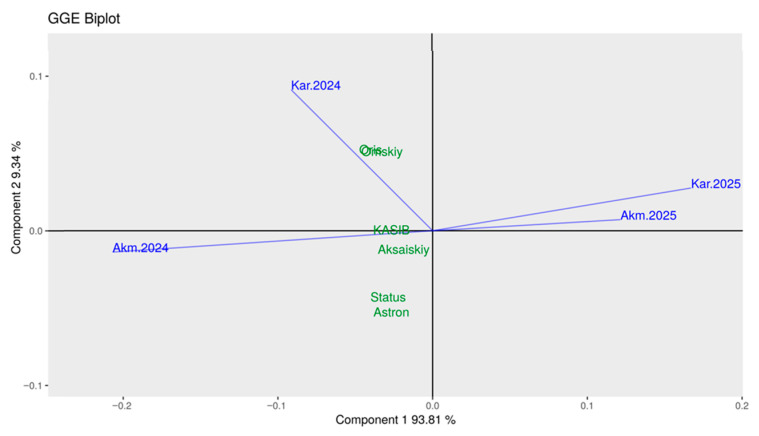
GGE biplot, Global-centred (E + G + GE) for seed yield of six major pea cultivars indicated in green, and two geographic locations, Akmola and Karaganda, in two years, indicated in blue with vector lines.

**Table 1 plants-14-03479-t001:** Statistical analysis of the comparison between genotyping of the *Le* gene and phenotyping for PH (**A**) and between the *Af* gene and leaf type (**B**) in all 66 pea plants studied. Differences between the two groups of *Le* genotypes and their PH phenotyping were estimated using a *t*-test (**A**), whereas Fisher’s Exact Test was employed to estimate differences between *Af* genotyping and leaf-type phenotype (**B**).

**(A)**						
**Genotype *Le***	**Fluorophore**	**Total Plants**	**Phenotype**, PH	**PH, Aver., cm**	**SE**	***t*-Test Between Groups**
*aa*	FAM	25	Short, semi-dwarf	56.3	3.4	*p* = 0.000000135; (*p* < 0.001)
*bb*	HEX	9	Medium PH	83.6	2.7	
		32	Tall			
**(B)**						
**Genotype *Af***	**Haplotype**	**Total Plants**	**Phenotype**, **Leaf Type**	**Total Plants**	**Differences**	**Fisher’s Exact Test**
*Afila*, WT	1–4	48	Regular	49	1	*p* = 1.000
*afila*, mut.	5–9	18	Semi-leafless	17	1	

**Table 2 plants-14-03479-t002:** Comparison of lodging index (LI), plant height (PH), and seed yield in six major pea cultivars in field trials in 2024 (**A**) and 2025 (**B**) in Karaganda region (Central Kazakhstan) and in Akmola region (Northern Kazakhstan). Relative yield class (RYC) was abbreviated as high (H), medium (M), and low (L), with significant differences between classes. Significant differences within columns were designated by different letters with probabilities of at least *p* < 0.05 using one-way ANOVA.

Cultivars	PH Class	Leaf Type	LI	PH, cm	Yield, kg/m^2^	RYC	LI	PH, cm	Yield, kg/m^2^	RYC
**(A)**			Akmola reg., 2024, early drought and sufficient moisture after				Karaganda reg., 2024, moderate early drought and sufficient moisture after			
Oris	Medium	Regular	4	80.2 ^b^	0.188 ^a^	H ^a^	5	56.5 ^c^	0.180 ^a^	H ^a^
Status	Short	Semi-leafless	5	65.9 ^c^	0.182 ^a^	H ^a^	5	36.9 ^d^	0.086 ^c^	L ^c^
Astronaute	Short	Semi-leafless	5	60.4 ^c^	0.177 ^b^	M ^b^	5	31.5 ^d^	0.076 ^c^	L ^c^
Omskiy neosip.	Tall	Regular	3	98.1 ^a^	0.174 ^b^	M ^b^	4	89.8 ^a^	0.176 ^a^	H ^a^
KASIB	Tall	Semi-leafless	3	95.0 ^a^	0.168 ^c^	L ^c^	5	69.9 ^b^	0.123 ^b^	M ^b^
Aksaiskiy usaty	Medium	Semi-leafless	4	78.1 ^b^	0.165 ^c^	L ^c^	5	58.2 ^c^	0.111 ^b^	M ^b^
*Average*			4	79.6	0.176		4.8	57.1	0.125	
**(B)**			Akmola reg., 2025, early drought and excessive rains after				Karaganda reg., 2025, moderate early drought and increased after			
Oris	Medium	Regular	3	76.8 ^b^	0.055 ^a^	H ^a^	4	59.2 ^b^	0.022 ^b^	M ^b^
Status	Short	Semi-leafless	4	69.4 ^c^	0.041 ^b^	M ^b^	5	42.7 ^c^	0.014 ^c^	L ^c^
Astronaute	Short	Semi-leafless	4	65.1 ^c^	0.042 ^b^	M ^b^	5	39.4 ^c^	0.012 ^c^	L ^c^
Omskiy neosip.	Tall	Regular	1	85.5 ^a^	0.025 ^d^	L ^c^	4	65.5 ^a^	0.032 ^a^	H ^a^
KASIB	Tall	Semi-leafless	2	85.8 ^a^	0.033 ^c^	L ^c^	4	62.3 ^ab^	0.032 ^a^	H ^a^
Aksaiskiy usaty	Medium	Semi-leafless	3	76.0 ^b^	0.054 ^a^	H ^a^	4	60.5 ^b^	0.023 ^b^	M ^b^
*Average*			2.8	76.4	0.042		4.3	54.9	0.023	

**Table 3 plants-14-03479-t003:** Distribution of 60 pea accessions with different plant heights (PHs) and leaf types for relative yield class (RYC) in plants grown in field trials in 2025 in Akmola (Northern Kazakhstan) and Karaganda (Central Kazakhstan) regions. RYC was abbreviated as high (H), medium (M), and low (L). Fisher’s Exact Test was used for comparison of RYC in plant phenotypes in two geographic regions. Significant differences are indicated by asterisks: * *p* < 0.05.

Plant Phenotype	RYC Akmola Reg., 2025			Total Plants	RYC Karaganda Reg., 2025			Total Plants	Fisher’s Exact Test
	H	M	L		H	M	L		
Tall PH with regular leaves	6	11	14	31	15	9	7	31	*p* = 0.0421 *
Medium PH with regular leaves	4	2	1	7	2	4	1	7	*p* = 0.7669
Low and medium PH and semi-leafless	7	11	4	22	4	6	12	22	*p* = 0.0481 *

## Data Availability

The original data presented in this study are included in the research paper and in Appendix A. Further inquiries can be directed to the corresponding authors.

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
