# Peer review of "Genotyping of Le and Af Haplotypes in Dry Pea (Pisum sativam L.) with Field Trials: Short and Semi-Leafless Plants Are Not Always Better in Kazakhstan"

_plants, 2025, doi:10.3390/plants14223479_

Round 1
Reviewer 1 Report
Comments and Suggestions for Authors
Plants-3930171: Genotyping of Le and Af Haplotypes in Dry Pea (Pisum sativam L.) with Field Trials: Short and Semi-Leafless Plants Are Not Always Better in Kazakhstan. This paper by Baurzhan Arinov et al. represents a meaningful topic on the lodging problem in pea plants. And it is essential to increasing the yield of pea under drought stress. Le (Stem length), Psat5g299720, for PH, and Af gene (Afila), Psat2g173360, for semi-leafless type with multiple tendrils, were further analyzed in the peas of different genotypes. This study suggested that earlier start and faster growth of taller plants with regular leaves can become much more important traits for better drought tolerance and seed production. However, this study did not investigated the mechanism of lodging problem in pea under normal conditions and drought stress. Thus, the research results and discussion of this paper are not in-depth. Considering the current status of the MS, it can not be published in Plants.
Author Response
Please see our responses in the attached file

Reviewer 2 Report
Comments and Suggestions for Authors
This manuscript presents a comprehensive study of a pea collection focusing on genes controlling plant height (the Le gene, Psat5g299720) and leaf modification (PsPALM1a, PsPALM1b, PsSA-RNA, PsNaOD1). The authors identified an SNP in the Psat5g299720 gene and demonstrated its association with plant height. Furthermore, they revealed various haplotypes of the PsPALM1a, PsPALM1b, PsSA-RNA, and PsNaOD1 genes, showing that the PsPALM1b gene influences leaf type development. Field phenotyping of the pea collection demonstrated that under drought conditions, short-stemmed forms are outperformed in terms of yield by medium and tall-stemmed genotypes. The research is comprehensive and systematic, and its findings could be valuable for practical breeding and the development of pea cultivars for conditions in Kazakhstan, as well as for fundamental studies on pea developmental genetics.
However, several remarks and questions regarding the work remain.
1) It is unclear whether the Psat5g299720-SNP1 and PsPALM1b (Psat2g173360) markers can be directly applied in marker-assisted selection.
2) From the article, it is not evident if there is a clear phenotypic difference (in height, yield) between the genotypic groups identified by the Psat5g299720-SNP1 and PsPALM1b (Psat2g173360) markers.
3) The abstract should be revised to include quantitative differences between the phenotypic and/or genotypic groups identified in the study.
4) lines 228-232: The grouping of phenotypes should be described in details rather than reffered to the published papers.
Kind regards,
Reviewer
Author Response

(The authors gave the same response as above.)

Reviewer 3 Report
Comments and Suggestions for Authors
The cited literature is mostly relevant and relatively recent, but the review lacks a clear thematic structure and critical synthesis. For example the References are presented as isolated findings from previous research without clear grouping (egs. by gene, trait or phenotype), and contradictory results in the literature are not sufficiently analyzed. To strengthen the scientific logic, authors should organize the literature more coherently and explicitly highlight the inconsistencies or gaps in knowledge that their study addresses. Inclusion of recent research on pheenotypic and genetic associations in pea would improve the background.
Results
The molecular genotyping results are presented descriptively, but lack statistical validation and integration with phenotypic data. In section 3.1, the distribution of Le genotypes is not statistically associated with plant height or lodging resistance, despite these being key hypotheses. Similarly, in 3.2, Af haplotypes are listed without explaining how they were derived or tested for phenotypic effects. The term “haplotype” may be inappropriate if not based on phased sequence data. These sections require stronger analytical treatment and clearer linkage to the study’s objectives.
Discussion
is primarily descriptive and lacks critical evaluation of the findings. Although the authors refer to existing literature, comparisons are superficial and lack specificity, absence of statistical associations between molecular and field data is not addressed, and limitations of the study are not acknowledged. Moreover, the discussion does not offer any novel insights or interpretations beyond confirming known patterns in plant height and lodging resistance. Stronger integration of genetic and phenotypic findings is needed to enhance the scientific value of this section.

Author Response
Please see our responses in the attached file.

Round 2
Reviewer 1 Report
Comments and Suggestions for Authors
At present, the revised MS has been greatly improved. I suggest it be published in Plants.
Author Response
We agree with Reviewer 1.
Reviewer 2 Report
Comments and Suggestions for Authors
Thank you for the revised version.
All my comments were addressed.
However, I could not find quantitative differences between groups of genotypes and
their phenotypes included in the Abstract. Please, check it.
Author Response
We provided additional information in Abstract addressing the comment of the Reviewer 2. Please ignore attached file here because it was uploaded for Reviewer 3.

Reviewer 3 Report
Comments and Suggestions for Authors
Please, see notes and suggestions in the paper.

Author Response
Please find our response in the attached 'pdf' file
